# Body side-specific control of motor activity during turning in a walking animal

**Matthias Gruhn\*, Philipp Rosenbaum, Till Bockemühl, Ansgar Büschges**

Department of Animal Physiology, Biocenter, University of Cologne, Cologne, Germany

**Abstract** Animals and humans need to move deftly and flexibly to adapt to environmental demands. Despite a large body of work on the neural control of walking in invertebrates and vertebrates alike, the mechanisms underlying the motor flexibility that is needed to adjust the motor behavior remain largely unknown. Here, we investigated optomotor-induced turning and the neuronal mechanisms underlying the differences between the leg movements of the two body sides in the stick insect *Carausius morosus*. We present data to show that the generation of turning kinematics in an insect are the combined result of descending unilateral commands that change the leg motor output via task-specific modifications in the processing of local sensory feedback as well as modification of the activity of local central pattern generating networks in a body-side-specific way. To our knowledge, this is the first study to demonstrate the specificity of such modifications in a defined motor task.

## Introduction

An animal's motor system faces complex demands in order to ensure proper function and survival. Such demands arise either from the intrinsic need to find food, or a mating partner, or the extrinsic needs to avoid predators or simply overcome obstacles. The investigation of the neural mechanisms underlying the generation of locomotion has unraveled considerable detail on how those neural networks that produce the basic motor outputs for swimming, flying and walking of invertebrates and vertebrates are organized and operate (*Grillner et al., 2008*; *Goulding, 2009*; *Lehmann, 2004*). At the level of the locomotor organs, a basic motor output is the result of the activity of local central pattern generating networks (CPGs), and its adjustment through sensory feedback from the locomotor organs (*Büschges and Gruhn, 2008*; *Pearson, 2008*). Particularly important for the control of the step cycle in this context is load (*Schmitz and Stein, 2000*; *Duysens et al., 2000*). The activity of the CPGs is initiated and modified depending upon the behavioral tasks the animal faces. Evidence suggests that descending signals from brain centers such as reticulospinal neurons in the vertebrate hind brain/basal ganglia (*Stephenson-Jones et al., 2011*) or the head ganglia of arthropods with the putative arthropod basal ganglia homolog (*Strausfeld and Hirth, 2013*), the central complex in the cerebral ganglion (*Strauss and Heisenberg, 1993*; *Guo and Ritzmann, 2013*; *Bidaye et al., 2014*; *Martin et al., 2015*) contribute to both the basic drive and its modulation (*Brocard et al., 2010*; *Shik et al., 1969*; *Martin et al., 2015*). How these descending signals influence the motor activity at the level of the local neural networks that control the locomotor organs, however, is only beginning to be understood, in particular for walking, the most ubiquitous form of locomotion in terrestrial animals (*Ridgel et al., 2007*; *Dyson et al., 2014*; *Martin et al., 2015*).

When changing course, all walking animals need to change the kinematics of each leg substantially in order to separate the activity between the leg and body segments and produce the appropriate locomotor movements on the two body sides (*Jander, 1985*; *Gruhn et al., 2009*; *Rivera et al., 2006*; *Dürr and Ebeling, 2005*; *Mu and Ritzmann, 2005*; *Jindrich and Full, 1999*; *Strauss and*

\*For correspondence: mgruhn@ uni-koeln.de

**Competing interests:** The authors declare that no competing interests exist.

**eLife digest** Walking along a curve or turning is a complex manoeuvre for the nervous system, as it must coordinate different leg movements on each side of the body. Rhythmic processes such as walking are controlled by networks of neurons called central pattern generators. The resulting movements can be adjusted by feedback from sense organs in response to environmental conditions. For example, sensory feedback that provides information about the load placed on each leg, allows the animal to control the duration of a stance. How the nerve cells, or neurons, involved in these processes work together to produce complex, flexible movements such as turning is largely unknown.

Previous work on how the brain negotiates turning movements has been carried out mostly in animals that swim or fly. To understand what happens during walking, Gruhn et al. monitored stick insects that walked in a curve on a slippery surface, and recorded the electrical activity within the animals' nervous system as they turned.

By comparing the activity of the nervous system on each side of the body while the insects walked a curve, Gruhn et al. found that the nervous system uses at least three different mechanisms to produce the different movements on the inside and outside. Firstly, the sensory feedback signals that communicate the load on the leg are processed in the legs on the outside of the curve to support forward steps, while they are processed on the inside legs to support forward, sideward, and backward steps. Secondly, the motor activity produced by the central pattern generator is modulated to be stronger for the muscle that moves the leg backward on the outside of the curve. At the same time, this activity is stronger for the muscle that moves the leg forward on the inside of the curve. Thirdly, signals from a front leg influence the movement of the other legs on the same side of the body. This influence is strong on the inside and weak on the outside of the curve.

Together or separately, these three mechanisms could provide the animal with the means to perform turns in all their different curvatures. Future work will need to work out exactly which local neurons process the signals sent from the brain to control movement.

*Heisenberg, 1990*; *Musienko et al., 2012*). In a six-legged curve walking insect, for example, outside legs push the animal, while inside legs pull the animal into the direction of the curve (*Jindrich and Full, 1999*; *Dürr and Ebeling, 2005*; *Gruhn et al., 2009*), sometimes even reversing the stepping direction in single inside legs (*Gruhn et al., 2009*; *Cruse et al., 2009*). Currently, little insight exists, into how the nervous system generates such motor flexibility (*Guo and Ritzmann, 2013*; *Huang et al., 2013*; *Hellekes et al., 2012*; *Ridgel et al., 2007*; *Fagerstedt et al., 2001*; *McClellan, 1984*; *Martin et al., 2015*). It seems clear that sensory input from vestibular/antennal or optical areas is processed in the brains of vertebrates and insects alike, and transformed into descending neuronal signals that induce changes in the activity of the local networks which produce the ongoing motor patterns (*Guo and Ritzmann, 2013*; *Mu and Ritzmann, 2008a*; *Ridgel et al., 2007*; *Strauss and Heisenberg, 1993*; *Huang et al., 2013*; *Musienko et al., 2012*; *McClellan, 1984*; *Martin et al., 2015*). The nature of these changes that generate body-side-specific motor output, however, is still largely unknown.

In the present study we have therefore investigated the neural mechanisms underlying the generation of the motor output during curve walking in an insect. We first analyzed stepping activity as well as the activity and timing of leg muscles in tethered, intact animals, walking freely on a slippery surface during optomotor-induced curved walking. In a next step, we investigated the underlying mechanisms by studying the local processing of load signals on the respective inside and outside of the curve. Finally, we investigated the centrally generated local motor output in a deafferented preparation and tested if the observed behavioral changes involved also changes in the activity of local central pattern generating networks.

Our results show that the leg motor output in optomotor induced curve walking insects results from body-side-specific modulation of the local, segmental processing of sensory feedback and the modulation of the activity of CPGs that control the individual leg joints.

# Results

## Turning of the intact tethered animal

During curved walking in insects, outside legs usually perform slightly modified forward steps, while the inside leg pulls the animal into the direction of the curve, and may produce forward, sideward and backward steps independent of the conditions under which walking is studied (*Mu and Ritzmann, 2005*; *Jindrich and Full, 1999*; *Szczecinski et al., 2014*; *Strauss and Heisenberg, 1990*; *Dürr and Ebeling, 2005*; *Gruhn et al., 2009*; *2011*). To understand the neuronal basis for these differences, we first studied turning of intact stick insects on a slippery surface (*Gruhn et al., 2006*). Despite some differences to walking on solid ground that will be discussed below, this arrangement excludes inter-leg influences from mechanical coupling through the ground, and a passive entrainment of leg activity through that of neighboring legs.

Quantitative analysis of 14 inside and outside curve walking sequences from 10 intact animals showed that the period of front leg (FL) inside steps ($P_{in}$=0.83s; SD=0.21) between animals was significantly shorter than that of FL outside steps ($P_{out}$=1.21s; SD=0.37) in 11 out of 14 sequences (p<0.05). Likewise, the average period of middle leg (ML) inside steps ($P_{in}$=0.76s; SD=0.15) was significantly shorter than that of ML outside steps ($P_{out}$=1.13s; SD=0.28) (p<0.01, Figure *Figure 1a*). Neither the

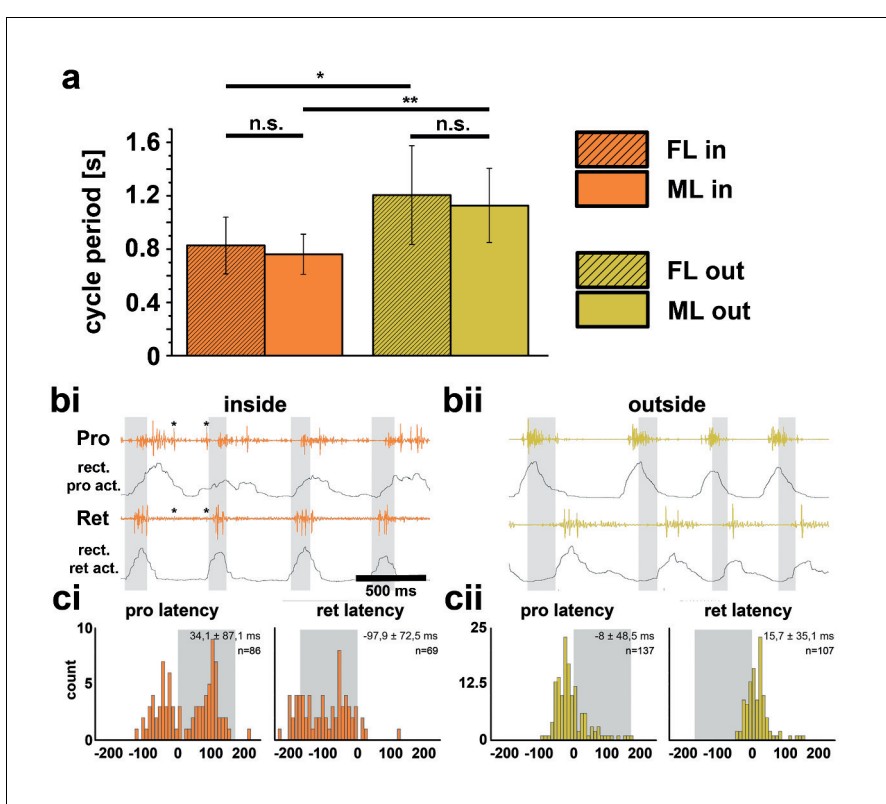

**Figure 1.** Differences between inside and outside middle leg (ML) activity in the intact tethered stick insect that is turning on the slippery surface. (a) Significant difference between the periods of inside (orange) and outside (yellow) turning of the front (striped) and middle (solid) leg, and no significant differences between the periods of front and middle leg stepping on either the inside or the outside; asterisks mark significance levels: * p<0.05, ** p>0.01. (bi–ii) Original and rectified EMG traces of *protractor* (Pro) and *retractor coxae* (Ret) with stance and swing (grey shaded bars) monitored electrically. The example in bi (orange) shows examples of backward steps during inside stepping with the retractor being active mostly in swing. Asterisks (*) mark very weak crosstalk between the two channels; bii (yellow), original and rectified EMG traces of the same muscles during outside steps. Here the retractor is always active in stance. ci and cii. Protractor and Retractor latency of the first EMG spike during inside (ci, orange) and outside (cii, yellow) steps from N=5 animals. Note the bimodal distribution during inside steps due to the occurrence of forward and backward steps.

stepping periods of the inside, nor those of the outside FLs and MLs were significantly different from one another ($p_{in}$ =0.43s, $p_{out}$= 0.6s, *Figure 1a*). We also looked for changes in motor output at the level of the muscle activity between the inside and outside legs. The only clear difference in the average EMG activity was observed in the most proximal leg joint, the thorax-coxa-joint, through which movements of a leg along the body axis are controlled. During inside stepping, the muscles *retractor coxae* and its antagonist, the *protractor coxae*, both served as either stance or swing muscles. Figures 1bi and bii show examples of ML *pro-* and *retractor coxae* EMG activity during inside and outside walking, and the averages of the rectified and smoothed EMG activity from 182 inside (*Figure 1ci*) and 204 outside steps (*Figure 1cii*) from five animals. Interestingly, the protractor activation in this case often also overlapped with that of the retractor, which suggests co-contraction and matches sideward steps observed during inside stepping. In addition, depressor activity on the outside was slightly prolonged (data not shown). Altogether, these findings corroborate the relative independence of the CPGs for each joint, and between hemiganglia that has been long known (*Büschges et al., 1995*; *Dürr and Ebeling, 2005*).

## Contribution of load feedback to turning kinematics

How do these drastic differences in leg kinematics, and the activity in the thorax-coxa leg joint, come about? We examined whether sensory feedback encoding load, which is known to be crucial for the control of stance motor activity during forward and backward walking, is controlling leg kinematics during turns (*Akay et al., 2007*).

During outside stepping sequences, load stimuli either initiated or enhanced retractor activity (*Figure 2*; 359 stimuli during outside walking in 13 animals). The timing of the load stimuli with respect to the front leg steps was distributed uniformly around FL stepping activity (*Figure 2c* and *Figure 2—figure supplement 1*). Processing of loading signals on the outside of curve walking animals is thus similar to straight stepping and ensures leg retraction throughout stance (*Akay et al., 2007*; *Borgmann et al., 2009*). On the inside, however, the response to a loading stimulus was markedly different. *Figure 3a* shows an example of mesothoracic protractor and retractor MN activity during application of load stimuli while the animal was standing, and during inside steps of the FL ipsilateral to the recording site.

Upon start of FL inside stepping, the mesothoracic protractor MNs showed an increase in activity. In contrast to the standing animal, however, no stereotypical activation of retractor activity was observed upon load stimuli (n=617). Instead, load stimuli either started or terminated retractor, and *vice versa*, protractor activity. Activation of the retractor MNs in variable strength compared to control upon load stimuli was observed in all 13 animals (16–100% of the stimuli/animal), termination of retractor, and activation of protractor MNs was observed in 54% of the cases (7 out of 13 animals, 2–25% of the stimuli). In addition, in 10 out of 13 animals, and on average upon 18.8% of the stimuli (SD=15.2), load stimuli did neither elicit retractor nor protractor activation. This is depicted for one animal in *Figure 3*, including PSTH analysis of the responses to CS stimuli during standing and during inside turning sequences (*Figure 3bi–iii*). Timing of the load stimuli with respect to the front leg steps was distributed uniformly throughout the FL step cycle. We also tested whether the strength of the retractor activation was dependent on the timing of the CS stimulus during the FL step cycle. This was not the case, as shown in the plot in *Figure 3c* for the example animal shown in *Figure 3a*, nor for all other animals (*Figure 3—figure supplement 1*). We equally tested whether either activation of the protractor motor activity or the lack of a response occurred during a preferred phase of the step cycle. No phase preference was found for the activation of protractor activity during FL step cycle (*Figure 3d*). However, we found a significantly increased likelihood for failure to elicit a response to a CS stimulus at 0.89 of the FL step cycle (p=0.028) although this phenomenon could occur during all phases of the FL step cycle (*Figure 3e*). In summary, these results clearly show side-specificity in the processing of local load signals during inside curve walking, modified from that during forward and outward stepping, such that the stereotypical influence of load feedback is no longer present.

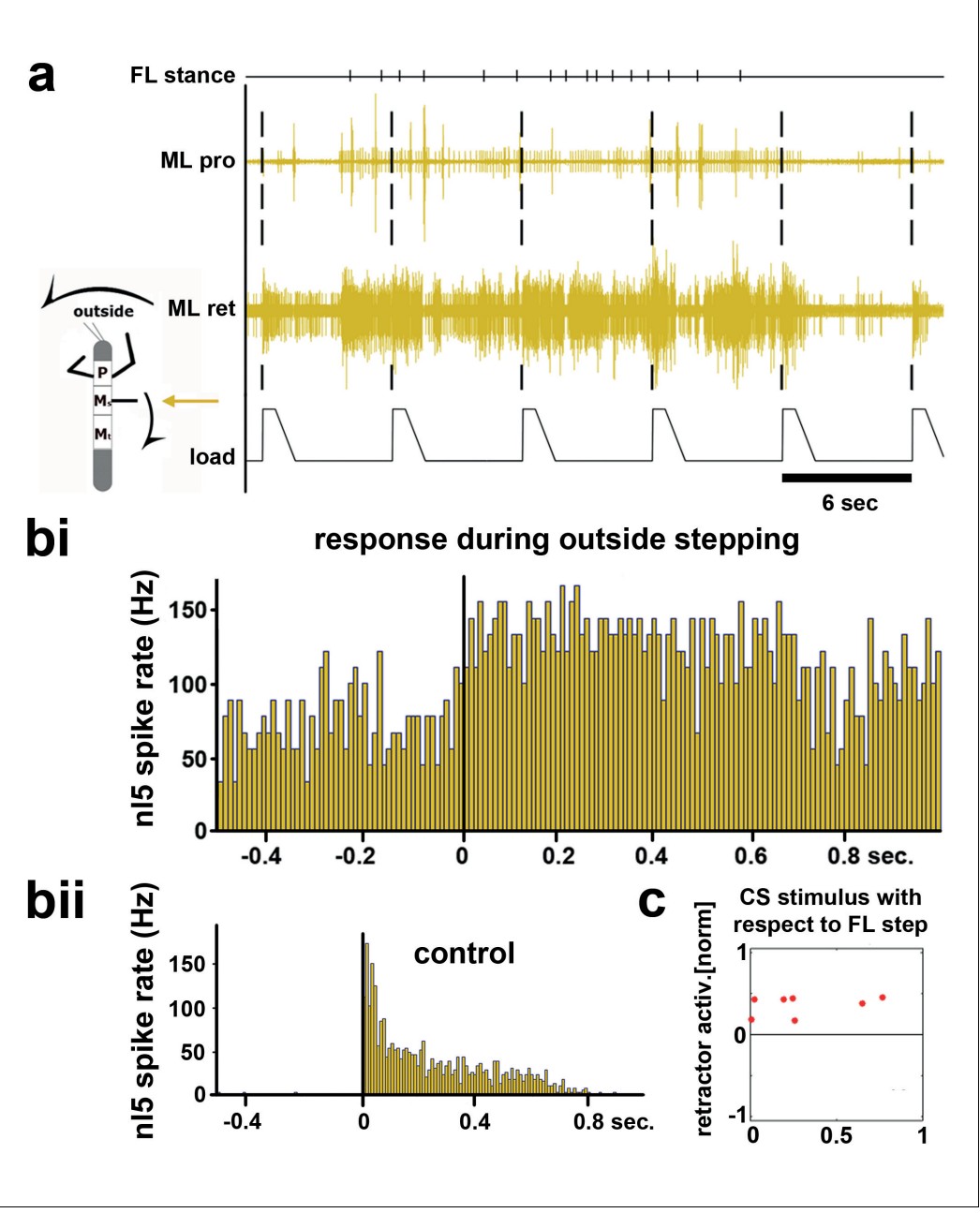

**Figure 2.** Influence of local load stimuli on mesothoracic motor activity during outside turning. (a) Mesothoracic pro- and retractor activity (2[nd] and 3[rd] row, resp.) before, during and after a front leg outside walking sequence (FL stance begin marked in top row event channel) and simultaneous application of load stimuli to the mesothoracic leg stump (bottom trace). (bi) Peristimulus time histogram (PSTH) showing the increase in mesothoracic retractor MN activity in response to local load stimuli during FL outside stepping; note the increased background activity. (bii) PSTH of retractor activity timed to the begin of the CS stimulus ramp in the quiescent animal (control). (c) Increased retractor MN activity upon CS stimuli did not depend on the phase of the CS stimulus within the FL step cycle during outside steps for the animal shown in a and b; 0 on the y axis marks the normalized baseline response in the 100ms before the CS stimulus. *Figure 2—figure supplement 1* shows the same data for all 13 animals. FL: front leg, ML: middle leg, pro: protractor, ret: retractor.

The following figure supplement is available for figure 2:

**Figure supplement 1.** Distribution of response amplitudes of retractor MNs to CS stimuli throughout the FL step cycle during outside steps from all 13 animals, normalized to the activity in the 100ms preceding the CS stimulus.

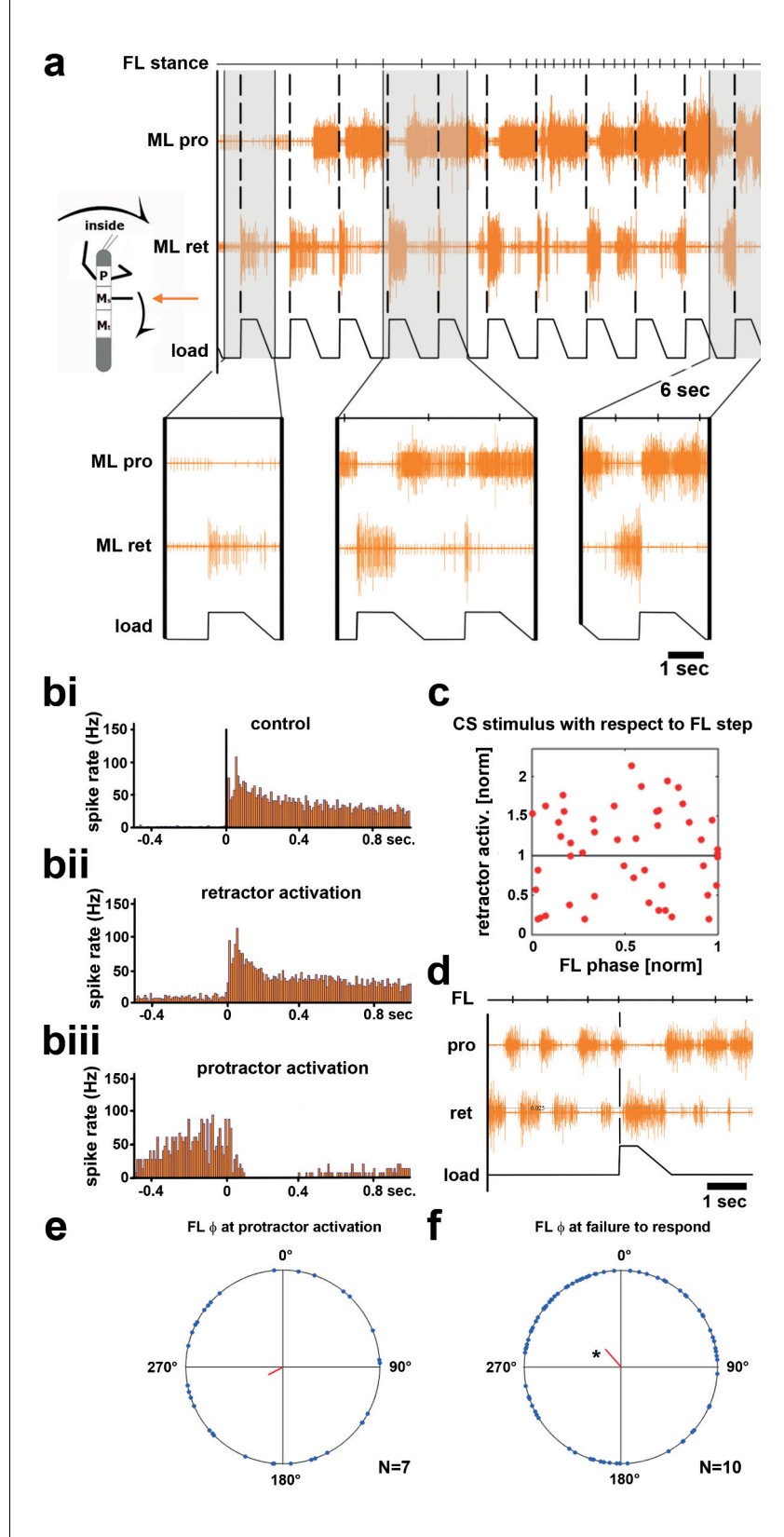

**Figure 3.** Influence of local load stimuli on mesothoracic motor activity during inside turning. (**a**) Mesothoracic pro- and retractor activity (2nd and 3rd row, resp.) before and during a front leg inside walking sequence (front leg

*Figure 3 continued on next page*

*Figure 3 continued*

[FL] stance begin marked in top row event channel), and simultaneous application of load stimuli to the mesothoracic leg stump (bottom trace). Four example responses (grey shaded areas) with expanded time resolution are shown below, the first one during quiescence, the other three during inside steps of the ipsilateral FL. (**bi–iii**) PSTHs showing the forms of mesothoracic retractor MN response to local load stimuli during FL inside stepping: **bi**, peristimulus time histogram (PSTH) of retractor activity timed to the begin of the CS stimulus ramp in the quiescent animal. **bii**, retractor activation upon load stimulus; **biii**, termination of retractor activity (activation of protractor activity is not shown as PSTH). (**c**) Distribution of response strength of retractor MNs to CS stimuli throughout the FL step cycle during inside steps with retractor activation from the animal shown in **a**; note that there is no phase preference for either an increase or a decrease in retractor activation compared to controls, this was also found for all other animals (*Figure 3—figure supplement 1*). (**d**) Example for no response to a loading stimulus. (**e**) Phase plot showing the distribution of CS stimuli in the FL step cycle that lead to protractor activation from N=7 animals; no significant phase preference was detectable. (**f**) Phase plot showing the distribution of CS stimuli in the FL step cycle that did cause neither re- nor protractor activation from N=10 animals; a significant phase preference was detectable at 0.89. FL: front leg, ML: middle leg, pro: protractor, ret: retractor.

The following figure supplement is available for figure 3:

**Figure supplement 1.** Distribution of response amplitudes of retractor MNs to CS stimuli throughout the FL step cycle during inside steps with retractor activation from all 13 animals (note that 9a and 9b constitute data from two files from one animal).

## Motor output during turning

In light of the significant role sensory feedback plays in the generation of the motor output for stepping in the stick insect, the results presented above can offer some explanation for observed differences in stance kinematics in the curve stepping animal (*Gruhn et al., 2009*; *Dürr and Ebeling, 2005*). Nevertheless, the question as to the mechanism behind the differences in cycle periods of outside and inside legs upon removal of mechanical coupling remains. We therefore checked for potential differences in the activity of those local central networks that generate alternating activity in the coxal protractor and retractor motoneurons (*Büschges et al., 1995*). To do so, we analyzed their motor output in the deafferented mesothoracic ganglion during curve walking. *Figure 4* shows mesothoracic protractor and retractor MN activity during inside (a–c), and outside turning (d–f) of the FL ipsilateral to the recording site. When the FL performed inside steps, protractor activity in the mesothorax was increased over retractor activity in all animals (N=17), and alternating activity between the two MN pools was observed (*Figure 4a*). The rhythmic activity of the mesothoracic coxal motoneurons was tightly correlated and largely in phase with the steps of the ipsilateral FL, as previously described for straight walking of the FL on a treadmill (*Borgmann et al., 2007*) (*Figure 4bi–ii*). Protractor activity was strongest at about 270° of the step cycle in the FL in about 60% of the experiments (N=17), retractor activity at about 90° in 86% of the experiments (N=14) (*Figure 4ci–ii*). During very few inside turning sequences protractor activity became almost tonic, while retractor activity was almost absent.

In contrast, during outside turns of the ipsilateral FL, protractor activity was strongly reduced compared to retractor activity, and the activity of both motor neuron pools was no longer alternating, but generally tonic ($N_{pro}$=17, $N_{ret}$=15; *Figure 4d*). Consequently, no significant phase coupling to the FL steps was present in the majority of cases, that is in about 76% of the animals for the protractor, and in 80% of the animals for the retractor (*Figure 4e–f*). In summary, coxal motoneuron activity on the inside of the curve walking animal was rhythmic and alternating, coupled to front leg stepping, while motoneuron activity on the outside was tonic and mostly limited to the retractor motoneurons.

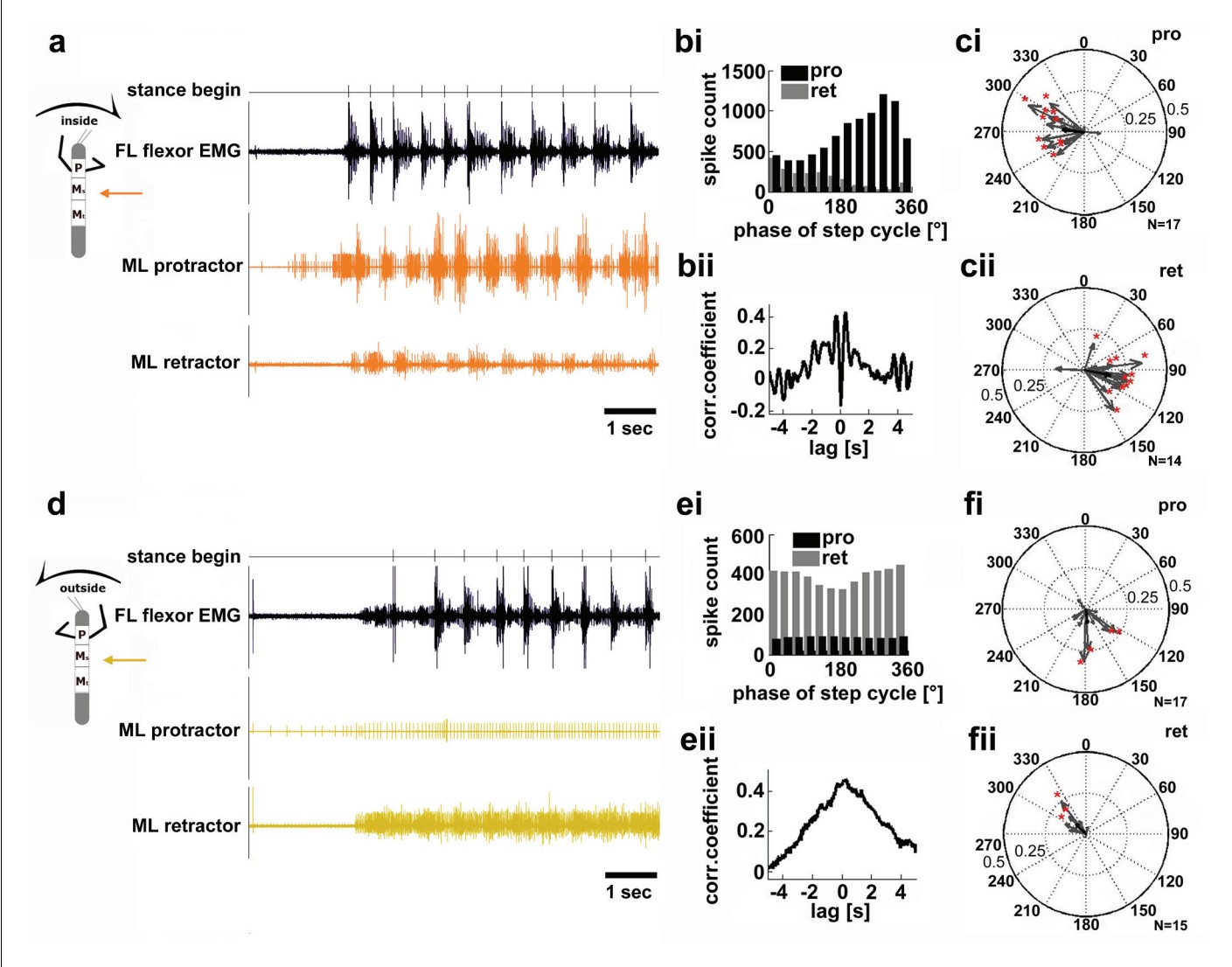

**Figure 4.** Motor output of the *protractor* and *retractor coxae* motor neuron pools in the deafferented mesothoracic ganglion during front leg inside and outside stepping. (**a**) Flexor EMG recording of the front leg (FL, top trace, black) together with simultaneous recording of ipsilateral mesothoracic protractor and retractor nerve activity (2nd and 3rd row, resp., orange) during an inside stepping sequence. (**bi**) Phase histogram of protractor and retractor activity with respect to the step cycle of the FL. bii. Cross correlation function of protractor and retractor activity. (**c**) Polar plots of directionality vectors with respect to the step cycle of the ipsilateral FL for protractor (**ci**) and retractor (**cii**) from N=17 and N=14 animals, resp.. (**d**) Flexor EMG recording of the FL (top trace, black) together with simultaneous recording of ipsilateral mesothoracic protractor and retractor nerve activity (2nd and 3rd row, resp., yellow) during an outside stepping sequence from the same animal as in a. (**ei**) Phase histogram of protractor and retractor activity with respect to the step cycle of the FL. (**eii**) Cross correlation function of protractor and retractor activity. (**f**) Polar plots of directionality vectors with respect to the step cycle of the ipsilateral FL for protractor (**fi**) and retractor (**fii**) from N=17 and N=15 animals, resp..

## Involvement of the local CPGs

The above results gave rise to the question whether the observed side-specific changes in coxal motoneuron activity were the result of influences on the activity of the local hemiganglionic CPGs that drive both MN pools (*Büschges et al., 1995*). We therefore isolated the mesothoracic ganglion in a split-bath preparation, and, after recording mesothoracic pro- and retractor activity in control conditions (*Figure 5ai, 6ai*), activated the local CPGs in the mesothoracic ganglion pharmacologically through superfusion with the muscarinic agonist pilocarpine (*Büschges, 1995*; *Borgmann et al., 2009*). This elicited slow, alternating motor activity in the standing animal (*Figure 5aii, 6aii, aiii*). During subsequent inside steps of the ipsilateral FL, protractor activity increased, and the

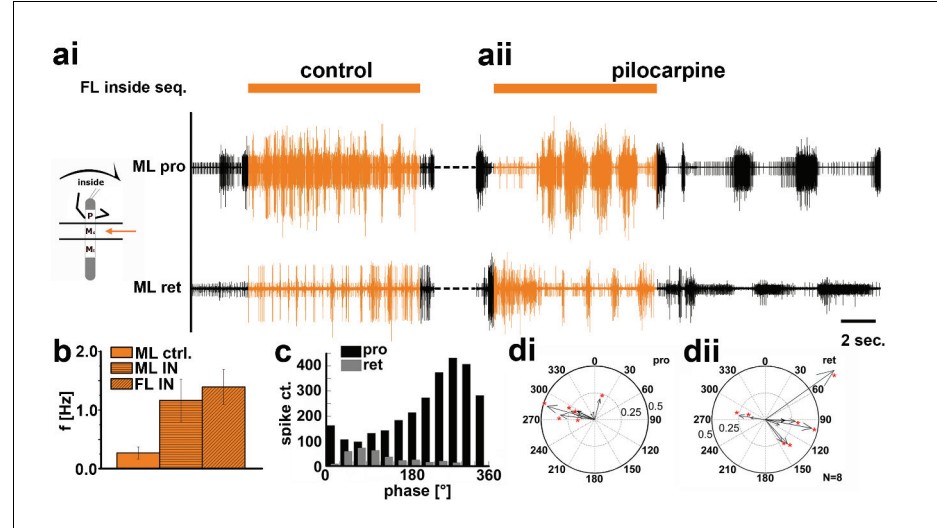

**Figure 5.** Motor output of the *protractor* and *retractor coxae* motor neuron pools in the deafferented control, and while the mesothoracic ganglion was superfused with a pilocarpine solution during front leg (FL) inside stepping in split-bath configuration. (a) Traces show the time with and without FL activity (top trace) together with simultaneous mesothoracic protractor and retractor nerve recordings (2nd and 3th row, resp.) ipsilateral to the FL, in control conditions (ai) and with pilocarpine on the mesothoracic ganglion (aii) of the same animal. (b) frequency of the mesothoracic pilocarpine rhythm in the quiescent animal (clear bar), and during inside steps of the FL (horizontally shaded bar) from N=11 animals. In addition, diagonally shaded bar shows the frequency of the stepping FL from the eight animals where the FL EMG was recorded. In all 11 cases the frequency of the pilocarpine rhythm during stepping sequences was significantly (p<0.001) increased over that in the quiescent animal. (c) Phase histogram of protractor and retractor activity from the same animal with respect to the step cycle of the FL during pilocarpine superfusion. (d) Polar plots with directionality vectors of protractor (di) and retractor (dii) with respect to the step cycle of the ipsilateral FL when the mesothoracic ganglion was superfused with pilocarpine (N=8 animals).

The following figure supplement is available for figure 5:

**Figure supplement 1.** Effect of unilateral transsection of the connective between the pro- and mesothoracic ganglion during inside stepping of the ipsilateral front leg (FL).

pilocarpine-induced rhythm sped up significantly in all experiments from an average 0.27Hz (SD=0.1) to 1.16 Hz (SD=0.36; N=11, p<0.001; *Figure 5b*). In about two thirds of the cases (62.5%; N=8) the frequency of the rhythmic mesothoracic MN activity was not significantly different from that of the stepping FL. Likewise, phase-coupling of mesothoracic MN activity to FL stepping was present, with protractor activity peaking around 270°, and retractor activity around 90° of the step cycle (*Figure 5c,d*).

In contrast, during outside turns of the FL, similar to the control condition without pilocarpine, retractor activity in the mesothoracic segment was strongly increased (*Figure 6ai*). No increase in mean frequency of the protractor/retractor MN activity was observed (N=8; $F_{ctrl}$=0.27, SD= 0.12, $F_{out}$=0.33, SD= 0.16), and the frequency was always significantly lower than that of the stepping FL (N=5; *Figure 6b*). In individual walking sequences, the frequency of coxal motoneuron activity even appeared to be locked in retractor phase (*Figure 6aii and aiii*). No systematic phase coupling to the outside stepping FL was present, again similar to controls (*Figure 6c,d*). In summary, our results thus show that side-specific changes in mesothoracic motor activity occur during turning that result from local changes in CPG activity. These changes in local motor activity appear to be influenced by FL stepping on the inside but not the outside.

To test whether these side-specific modifications of the mesothoracic motor output were based either on descending unilateral drive, through the corresponding connective, or bilateral drive through both connectives, we lesioned the connective between the pro- and mesothoracic ganglia ipsilateral to the recording site. This lesion abolished all of the above described modifications of MN

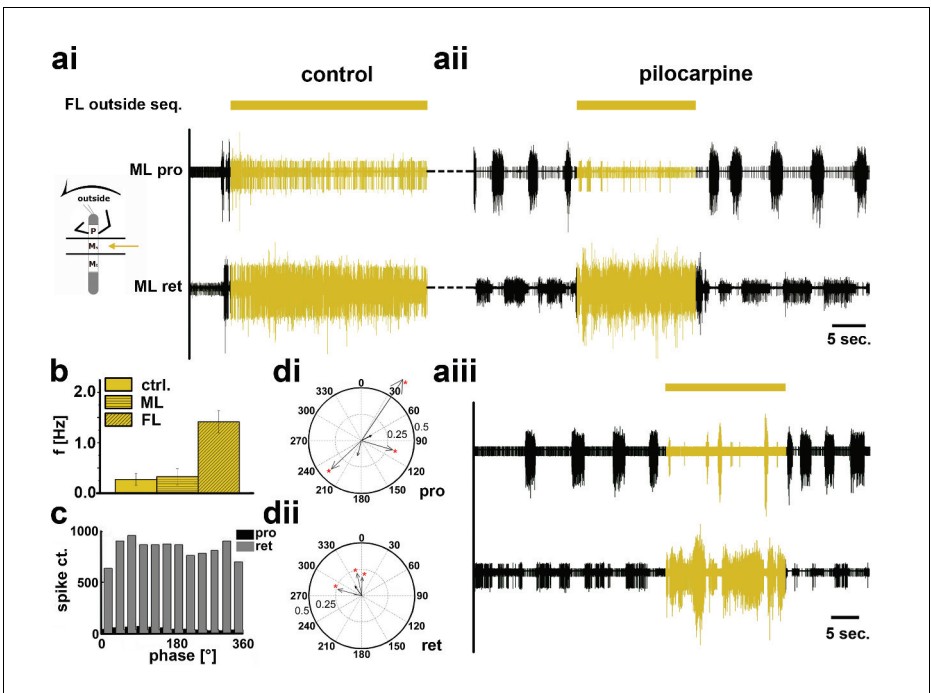

**Figure 6.** Motor output of the *protractor* and *retractor coxae* motor neuron pools in the deafferented control and when the mesothoracic ganglion was superfused with a pilocarpine solution during front leg (FL) outside stepping in split-bath configuration. (**a**) Traces show the time with and without FL activity (top trace) together with simultaneous mesothoracic protractor and retractor nerve recordings (2nd and 3th row, resp.) ipsilateral to the FL, in control conditions (**ai**) and with pilocarpine on the mesothoracic ganglion of the same animal (**aii**), and in a 2nd animal (**aiii**). (**b**) Frequency of the mesothoracic pilocarpine rhythm in the quiescent animal (clear bar) and during inside steps of the FL (horizontally shaded bar) from N=8 animals. In addition, the diagonally shaded bar shows the mean FL stepping frequency from the five animals where the EMG was recorded. Note that except for 2 animals the frequency of the pilocarpine rhythm during stepping sequences was not significantly different from control in the quiescent animal. (**c**) Phase histogram of protractor and retractor activity with respect to the step cycle of the front leg during pilocarpine superfusion from the animal in aii. (**d**) Polar plots with directionality vectors of protractor (**di**) and retractor (**dii**) with respect to the step cycle of the ipsilateral FL when the mesothoracic ganglion was superfused with pilocarpine (N=5 animals).

activity during curve stepping (*Figure 5—figure supplement 1*, N=4), incl. the influence on the activity of the segmental CPGs in the split-bath configuration (N=2), while the contralateral activity on the intact side remained unaffected. These findings demonstrate that the modifications in mesothoracic motor activity observed during FL curve stepping are the result of unilateral information flow through the ipsilateral connectives.

## Discussion

Using the stick insect, we investigated the neural basis for the generation of the motor output for curve walking. We focused on how local, thoracic information processing contributes to produce the observed differences in the motor output of the two sides of the body during optomotor induced turning. We have shown that animals walking on a surface with reduced mechanical coupling generate higher stepping frequencies in inside compared to outside legs. *Protractor* and *retractor coxae* muscles show major differences in activation during both conditions. Depending on the function of the leg as inside or outside leg, processing of local load feedback and the activity of the segmental CPG for the coxal motoneurons of the middle legs (ML) are modified in a task-specific manner. We conclude that these body-side-specific changes participate in altering leg kinematics during optomotor-induced turning. A schematic summary of the results is given in *Figure 7*.

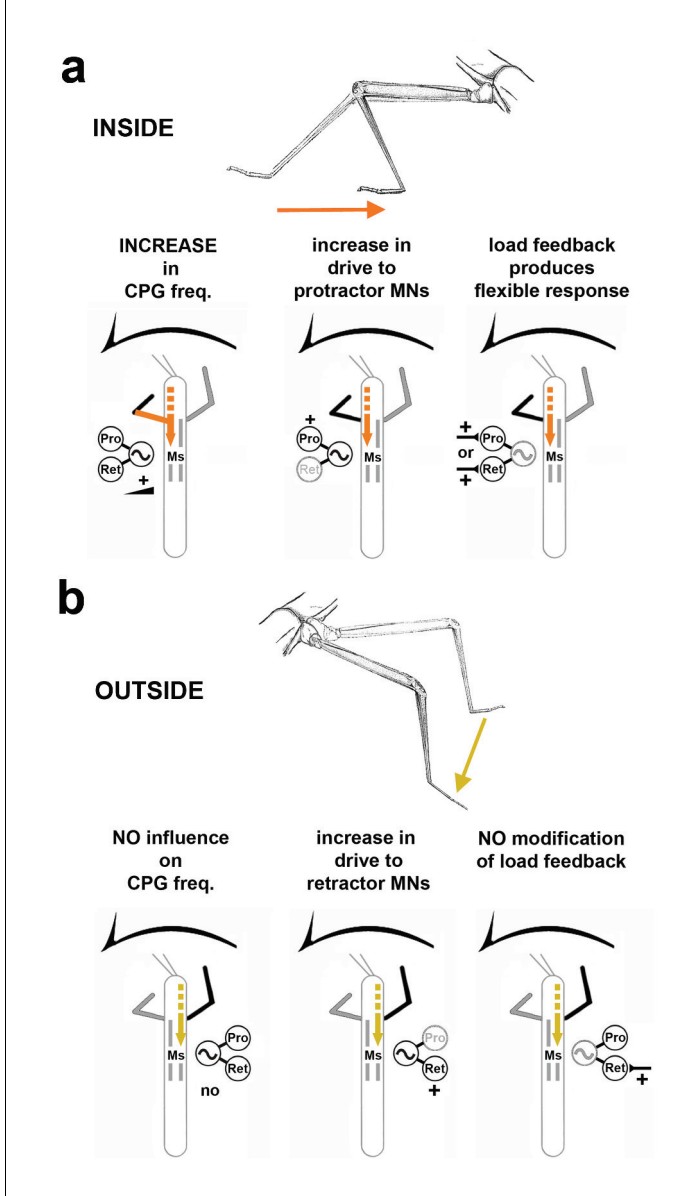

**Figure 7.** Summary of the observed effects on mesothoracic motor output and potential sources for the influences. (a) Effects observed in the mesothorax ipsilateral to the *INSIDE* front leg. (b) effects observed in the mesothorax ipsilateral to the *OUTSIDE* front leg. Note that solid lines mark demonstrated origin through lesion experiments, while broken lines mark suggested probable origin from higher order brain regions in the head, as discussed.

## Different leg kinematics and motor output on the two body sides of a curve stepping insect

Previous studies on curve walking reported varying degrees of decoupling and asymmetry between the limbs on the inside and outside of a curve (cat: (*Musienko et al., 2012*); turtle: (*Rivera et al., 2006*), stick insect: (*Jander, 1985*; *Dürr and Ebeling, 2005*; *Gruhn et al., 2009*); cockroach: (*Jindrich and Full, 1999*; *Mu and Ritzmann, 2005*); crayfish: (*Cruse and Saavedra, 1996*); *Drosophila*: (*Strauss and Heisenberg, 1990*)). The success of the turning motor behavior can either be deduced from changes in the stride length (*Strauss and Heisenberg, 1990*) or from the sideward or backward steps or strokes of the inside leg (*Gruhn et al., 2009*; *Strauss and Heisenberg, 1990*; *Mu and Ritzmann, 2005*; *Rivera et al., 2006*; *Szczecinski et al., 2014*). In the stick insect,

the ML movement on the inside is characterized by a flexion around the femur-tibia joint, reduced angle of movement in the thorax-coxa joint, and the occurrence of forward, sideward and backward steps in immediate succession. In contrast, outside MLs only show an increased front to back angle of movement, and reduced flexion in the femur tibia joint both under conditions of turning on normal substrate and during turns on the slippery surface (*Dürr and Ebeling, 2005*; *Gruhn et al., 2009-, 2011*). Most studies on turning of arthropods have been performed on air cushioned balls, and also reported a decrease in stepping frequency of the inside compared to the outside legs (*Zolotov et al., 1975*; *Jander, 1985*), sometimes no change in frequencies (*Cruse and Saavedra, 1996*).

These reports differ from our observation that the average cycle period of inside steps was significantly shorter than that of outside steps. However, all of the studies cited above investigated curve walking while the individual legs of the animals were mechanically coupled through the ground, and could not express their inherent cycle period. A plausible explanation for the difference to our findings and similar ones in the cockroach (*Mu and Ritzmann, 2005*) is that we studied curve walking on a slippery surface. During walking above greased plates, the mechanical coupling between the legs, and passive influences caused by ground contact are reduced or even abolished. Therefore, turning on the slippery surface must be considered a special case, in that the traction for the legs and the resistance the legs experience is reduced. In addition, stepping is not affected by passive displacement through the movements of the other legs on the ground via the mechanical coupling. Hence, the resulting sensory feedback is altered from walking on normal ground. Interestingly, turning kinematics between slippery surface and air-cushioned ball are still remarkably similar for both cockroach and stick insect (*Tryba and Ritzmann, 2000*; *Gruhn et al., 2009*; *Dürr and Ebeling, 2005*; *Szczecinski et al., 2014*). Because of these differences, walking on the slippery surface allows one to draw conclusions about the centrally generated motor output of the nervous system. The significantly higher stepping frequency observed on the inside compared to the outside under these conditions may therefore reflect exactly this changed output during turning which, under more naturalistic conditions, is masked and altered by sensory feedback.

## Different processing of local load feedback on both sides of a curve stepping insect

How are the differences in stepping frequency and kinematics generated? Experimental data point towards two alternative ways for the neuronal control of turning, that are not necessarily mutually exclusive. In lamprey, curve swimming is achieved by increasing the descending excitatory drive unilaterally which ultimately leads to steering into the direction of the body side with the increased drive (*Fagerstedt et al., 2001*; *Grillner et al., 2008*). During terrestrial locomotion, however, the rigid coupling through ground contact exerts much stronger influences than mechanical coupling in fluids and locomotion involves more than trunk movement. This might explain why changes in sensory processing have been reported to form the basis of context-dependent changes in motor output in insects (*Akay et al., 2007*; *Hellekes et al., 2012*; *Mu and Ritzmann, 2008b*; *Martin et al., 2015*).

We have shown that local load feedback signals influence coxal MN activity depending on whether the load stimulus is applied to the ML stump ipsilateral to an outside or inside front leg (FL). Similar context-dependent processing of sensory feedback was reported before in vertebrates and insects (*Pearson and Collins, 1993*; *Akay et al., 2007*; *Hellekes et al., 2012*; *Mu and Ritzmann, 2008b*; *Martin et al., 2015*; *Mu and Ritzmann, 2008b*). Feedback about flexion of the stick insect tibia, signaled by the femoral chordotonal organ (fCO), leads to a reinforcement of flexion in inside legs, while no such effect is observed in outside legs (*Hellekes et al., 2012*). Similarly, feedback from ML campaniform sensilla (CS) of stick insects, elicited by a load-mimicking stimulus in the horizontal plane, like that applied in our experiments, was shown to trigger a switch from ML protractor to retractor activity during forward, and in the opposite direction during backward stepping of the FL on a treadwheel (*Akay et al., 2007*). All these changes in sensory processing could be depending on descending commands from the brain, as lesioning the connective anterior to the prothoracic ganglion (*Mu and Ritzmann, 2008b*) and stimulation of turn-inducing central complex (CX) neurons have been shown to alter similar reflexes elicited by stimulation of the chordotonal organ in cockroach (*Martin et al., 2015*).

Our results suggest that processing of a loading stimulus on the outside is similar to that during forward stepping, leading to retractor activation. Processing of the same stimulus on the inside was not found to be so stereotypic, ranging from activation of retractor to different degrees and termination of protractor activity, through activation of neither motor neuron pool, to protractor activation and termination of retractor activity (*Figure 7*). Except for a very weak phase dependence in the failure to respond to a CS stimulus to the final fourth of the FL step cycle, all other types of response to loading stimuli were not dependent of the FL step cycle. On the outside, the response matches the functional stance movement of the outside leg in the intact animal and, similar to straight walking, load in this context supports retraction of the leg (*Akay et al., 2007*). On the inside, the variability in the influence is in good agreement with the observations from turning intact animals, where inside ML steps show the full spectrum of movement directions from forward to backward steps both on slippery ground and under more natural conditions (*Dürr and Ebeling, 2005*; *Gruhn et al., 2009*; *Cruse et al., 2009*). A context-dependent influence of load feedback as part of a unilateral mechanism to generate behavioral adaptivity had been shown (*Akay et al., 2007*). From our results, it seems clear that graded input from yet unknown origin anterior to the mesothoracic ganglion during optomotor induced curve walking of the front legs locally modifies strength and sign of this load stimulus-dependent influence. At least two possible mechanisms for the processing of load feedback are conceivable. One is presynaptic inhibition (*Stein and Schmitz, 1999*) that could modify the gain of load feedback onto the neural networks controlling the thorax-coxa joint CPG. The existence of presynaptic modulation of sensory synapses in the CNS has previously been described in a number of invertebrate and vertebrate locomotor systems (*Clarac and Cattaert, 1996*; *Büschges and Wolf, 1999*; *Sirois et al., 2013*). Alternatively, the weighing of parallel pathways from the load sensors, the CS, to the premotor interneurons could be shifted and thereby either promote retractor or protractor activation, as previously shown for gain control of reflexes (*Driesang and Büschges, 1996*) and recently implemented in a simulation model for cockroach turning (*Szczecinski et al., 2014*).

Taken together, the turning related modifications of the sensory processing on the two body sides always support stance. The difference between the two sides is that on the inside, flexion of the tibia and the inward directed movement of the leg are prioritized through strong assisting input from the fCO (*Hellekes et al., 2012*). In this context, loading feedback is of lesser importance for determining forward or backward movement. In contrast, on the outside, processing of sensory feedback produces the opposite effect in that loading supports retraction to promote the front-aft movement of the leg.

## Body-side specific influences on ThC-CPG activity during curve stepping

In addition to altered processing of local load feedback, we showed that in the intact animal phasing of the subcoxal protractor and retractor muscles with respect to stance and swing is markedly different between inside and outside steps. In addition, the motor output in the deafferented mesothoracic motor control system is also modified in a side-specific manner during curve walking of the FLs: first, there is pronounced alternating activity between pro- and retractor MNs on the inside, and tonic activity on the outside. Second, there is a strong bias towards protractor activation on the inside, and towards retractor activation on the outside. Our results thus indicate that the activity of the local subcoxal CPGs is under the influence of descending input from more anterior ganglia to support the generation of curve walking kinematics (*Figure 7*). How can such an influence be mediated?

Up to now, there are only two animal neural networks for which there exists a well founded idea on how turning is realized on the local level. The first are the networks that generate swimming in lamprey (*Grillner et al., 2008*), and the other are the networks that generate flight in locust (*Rowell, 1993*). In lamprey, curve swimming is presently viewed to be mediated by a unilateral increase in descending excitatory drive from the brainstem onto the segmental CPGs which leads to a bias in the contraction strength of the segmental trunk muscles, and consequently to swimming towards the side with the increased drive (*Fagerstedt et al., 2001*; *Grillner et al., 2008*). In the locust, asymmetries in wing beat amplitude, phase shifts of wing beat between the sides, bending the abdomen and ruddering with the hind legs, all form part of steering maneuvers (*Rowell, 1993*). Especially, if the pronation angle in both wings on one side changes, turns are induced as a result of differences in lift and thrust between the two body sides (*Wolf, 1990*). These asymmetries are created by activation of thoracic interneurons that receive input from descending interneurons about changes in body

orientation. This appears to directly affect MN output on one side of the body (*Reichert, 1989*; *Rowell, 1993*). However, in contrast to lamprey and the stick insect, in the locust, local CPG activity is unaffected by this descending input (*Rowell, 1993*).

In the curve walking stick insect, it is quite conceivable that a unilateral mechanism not completely unlike that in curve swimming lamprey is at work, as severing one of the two connectives between pro- and mesothoracic ganglion only prevented turning-related changes in motor activity on the lesioned side and not on the other. The existence of a descending drive from anterior to the meso-thoracic ganglion to the pattern generating networks of the mesothoracic joints was shown previously in physiological experiments (*Westmark et al., 2009*; *Borgmann et al., 2007*), and successfully implemented in models for the stick insect (*Knops et al., 2013*; *Toth et al., 2012*) and also for cockroach walking and turning (*Szczecinski et al. 2014*). In contrast to lamprey steering, however, this drive appears to affect the segmental CPGs on each side of the stick insect. Moreover, the drive to the mesothorax during turning produces particularly strong activity in the respective opposite coxal MN pools in the homologous CPGs of the two sides. The above mentioned model hypothesized, and our results suggest an independent descending drive to the two body sides, which can independently strengthen or weaken the motor outputs to single leg muscles on each side of the animal (*Knops et al., 2013*; *Toth et al., 2012*; *Szczecinski et al., 2014*).

## Potential sources for the influences on the thoracic network during turning

Currently, little information is available on where the drive onto the mesothoracic segment that induces the modification of local network activity in the stick insect comes from. The coupling of meso-thoracic inside motor activity, clearly demonstrates a phasic interganglionic thoracic influence through the FL activity. However, this influence is not present on the outside. This suggests that higher order centers in the subesophageal and supraesophageal ganglia may play the decisive role in shaping thoracic motor activity and sensory processing. Evidence for higher order centers in the insect brain as a source of the descending drive come from studies on the central complex (CX) of the cerebral ganglion. Lesion and stimulation experiments in the cockroach have shown that this neuronal structure is necessary for proper turning (*Ridgel et al., 2007*; *Martin et al., 2015*). Fruit flies with mutations in the CX, show severe deficits in turning and start/stop maneuvers (*Strauss and Heisenberg, 1993*). By lesioning the connectives anterior to the prothoracic ganglion, in cockroach and locust, or stimulation of the CX, descending signals from the brain were also shown to be responsible for altering reflex sign or gain (*Mu and Ritzmann, 2008b*; *Knop et al., 2001*; *Martin et al., 2015*) as part of the generation of turning kinematics. The nature of the descending signals and the cellular pathways are still unknown. Assuming a similar source in the stick insect, alteration of motor activity and sensory processing in the turning stick insect could be the result of increased excitation following disinhibition on one side (*Roeder, 1937*) or, through crossed inhibition, on the contralateral side (*Gal and Libersat, 2006*), and direct inhibition from the cerebral ganglion (*Gal and Libersat, 2006*). Our data allow both mechanisms to play a role in turning behavior, as both, inside and outside mesothoracic pro-/retractor CPGs show increased activity, only in opposite MN pools. Recently, a neuron class was described in *Drosophila* that seems both necessary and sufficient to drive backward walking (*Bidaye et al., 2014*). Sustained activation of the so-called MDN neurons (two per cerebral ganglion hemisphere) allow the flies to reverse stepping direction. Their axons arborize in both hemispheres of the cerebral and subesophageal ganglia, and cross the midline to project to the contralateral thoracic ganglia. From the report, it appears that this crossing takes place in the ventral protocerebrum (*Bidaye et al., 2014*). Unilateral activation of similar neurons in the stick insect could provide means to promote turning.

In summary, our results indicate that motor output of single legs in the turning stick insect is under individual hemi-segmental influence by descending signals. We have shown that these signals to the mesothoracic ganglion act together to produce turning kinematics by at least three mechanisms: first, task-dependent processing of local load signals, modified on the inside, and unchanged on the outside; second, a shift in the weighting of motor output to the protractor on the inside, and the retractor on the outside, via modification of mesothoracic CPG activity; and third, task-dependent influence from the ipsilateral front leg which is strong on the inside and weak on the outside. Together or separately, in full strength or gradually, these three mechanisms could provide the animal with the means to perform optomotor induced turns in all their different curvatures. Previous

work suggests that the origin for the modulatory influence on mesothoracic CPG activity resides largely anterior to the prothoracic ganglion. Our results provide the first comprehensive evidence for the combined local action of different central neuronal mechanisms acting together at the level of the thoracic ganglia to drive this specific kind of behavior in a legged animal.

## Materials and methods

### Animals

All experiments were performed in a darkened Faraday cage at room temperature (22°–24°C) on adult female stick insects (*Carausius morosus*; Brunner), of 7.5 cm in length. Animals were raised on blackberry leaves fed *ad libitum* and kept at a 12 hr:12 hr light dark cycle.

### Experimental setup

In all experiments, animals walked on a 13.5 cm x 13.5 cm polished nickel-coated brass plate covered with a lubricant (95% glycerin, 5% saturated NaCl) as described earlier (*Gruhn et al., 2006*). This produced slipperiness and allowed recording of tarsal contact when desired (*Gruhn et al., 2006*). Animals were glued ventral side down on an 80 mm long and 3 mm wide balsa rod using dental cement (ProTempII, ESPE, Seefeld Germany) so that legs and head protruded from the rod, and all joints were unrestrained. Animal height above the substrate was adjustable, but was typically 10 mm. Walking was elicited as an optomotor response by projecting a striped pattern (pattern wave length 21°) onto two 13.5 cm diameter round glass screens placed at right angles to each other and at a 45°angle to the walking surface, approximately 6–7 cm away from the eyes of the animal (*Gruhn et al., 2006*). If the animal did not begin locomotion spontaneously, walking was elicited by light brush strokes to the abdomen. For experiments with two-legged animals, we induced autotomy of the meso- and metathoracic legs with a pair of forceps or cut the legs at the level of the coxa after recording from the intact animal (*Borgmann et al., 2007*). After that we allowed a minimum of 30 min for recovery.

### Optical recording and digital analysis of leg movements

Walking sequences were recorded from above with a high speed video camera (Marlin F-033C, Allied Vision Technologies, Stadtroda, Germany) at 100fps as described (*Gruhn et al., 2009*; *2011*). Legs were marked at the distal end of the tibia, using yellow fluorescent pigments as markers (catalogue #56150, Dr. Georg Kremer Farbmühle, Aichstetten, Germany) which were dissolved in two-component glue (ProTempII, ESPE, see above). During the recording, the animal was illuminated with blue LED arrays (12 V AC/DC, Conrad Electronic, Germany). A yellow filter in front of the camera lens was used to suppress the short wavelength of the activation light. Video files were analyzed using motion tracking software (WINanalyze, Vers.1.9, Mikromak service, Berlin, Germany). Stepping frequencies between inside and outside front and middle legs were determined using the times of touchdown (at the anterior extreme position, AEP) and lift-off (at the posterior extreme position, PEP), as well as the period between consecutive AEPs. These were verified visually by an additional mirror placed at an angle of 45° behind the animal. All inside and outside steps of complete 20–30 s stepping sequences were averaged in their respective groups and evaluated. The sample size for the inside and outside walks was N=10 animals, and 1–2 stepping sequences per animal (total of 14), the walks used for the analysis of front and middle leg stepping frequency were the same.

### Recording tarsal contact

Latencies of muscle activation were determined using the electronic tarsal touchdown signal of the ML as described (*Gruhn et al., 2006*). In brief, a square wave signal of 2–4 mV amplitude was generated with a pulse generator (Model MS501, electronics workshop, Zoological Institute, Cologne), and applied to the slippery surface and a lock-in amplifier (electronics workshop, Zoological Institute, Cologne) as a reference signal. A copper wire (49 μm outer diameter) with its insulation removed at the tip was tied around the tibia of the monitored leg, and connected to the lock-in amplifier with an alligator clip. The electrical resistance between the cuticle and copper wire was reduced with a drop of electrode cream (Marquette Hellige, Freiburg, Germany) placed at the area of contact,

allowing a 2–4 nA current to pass through tarsus and tibia. During stance, current flowed from the plate through tarsus and tibia into the copper wire. The digitized using an AD converter (Micro 1401k II, CED, Cambridge, UK) and Spike2 software (Vers. 5.05, CED, Cambridge, UK). Due to the low-pass filter properties of the lock-in amplifier and the gradual lift-off/touchdown of the tarsus, the signal was not exactly square. We used thresholds close to the transition point to define the timing of tarsal contact and manually checked each event. Touchdowns could be determined at a resolution of less than 1 ms. Lift-off transitions were less steep and more delayed because of delayed tearing of the lubricant from the tarsus due to capillary action and occasional upward movements of the leg during stance without complete lift-off. To have comparable lift-off times in all experiments, we always defined lift-off as the time point with the steepest ascending slope.

## Electrophysiological recordings

Muscle activity (*electromyogram*, EMG) was recorded as described (*Rosenbaum et al., 2010*), using two twisted, coated copper wires (40 µm outer diameter) placed in each muscle approximately 1 mm apart and held in place with dental cement (ProTempII, ESPE) or tissue adhesive (3 M Vetbond, St.Paul, MN). All recordings were differentially amplified. The EMG signal was pre-amplified 100 fold (electronics workshop, Zoological Institute, Cologne), band-pass filtered (100 Hz-2000 Hz), when necessary further amplified 10–1000 fold, and imported into Spike2 (Vers. 5.05, CED) through an AD converter (Micro 1401k II, CED). A reference electrode was placed in the abdomen of the animal. For the analysis of muscle latency in the inside or outside ML, in most experiments, two antagonistic joint muscles were recorded simultaneously. *Protractor coxae* and *retractor coxae* EMG's were recorded in the thorax, *depressor trochanteris* and *levator trochanteris* in the coxa, and *extensor tibiae* and *flexor tibiae* in the femur. In 2 experiments three muscles, in 3 experiments four, and in one all six muscles were recorded from, simultaneously. These experiments gave the same results as the others. For the experiments in the two-leg preparations, EMGs in the flexor tibiae muscle of each leg served as a reference for the step cycle of the front leg. For *Figure 1*, the EMG activity was placed in the reference frame of the electrically determined swing and stance phases.

Extracellular nerve recordings were performed as follows: all except the two front legs were amputated at mid-coxa (*Borgmann et al., 2007*). Animals were fixed with dental cement (Protemp II, ESPE) dorsal side up on a foam platform with the same width as the balsa stick for the intact animals, but for a small stretch along the thorax behind the front legs where the platform was 8mm wide to accommodate the gut. The dorsal side of the thorax was opened, the gut moved aside, and connective tissue carefully removed to expose the connectives, the meso- and metathoracic ganglia and respective leg nerves. The coxae of the amputated middle and hind legs were fixed with dental cement, and afferent feedback was prevented by severing all lateral nerves of the meso- and metathoracic ganglia, except nl2 and nl5 (*Marquardt, 1940*) of the mesothorax, which contain the axons of the mesothoracic coxal protractor and retractor motor neurons (MNs), respectively. The body cavity was filled with *Carausius* saline (*Weidler and Diecke, 1969*), except for in split-bath experiments (see below). Mesothoracic motor activity was recorded extracellularly from leg nerves nl2, and nl5 (*Borgmann et al., 2007*) on both sides of the animal, using monopolar hook electrodes (modified after (*Schmitz et al., 1991*). For the lesion experiments, the pro-to-mesothoracic connective under investigation was cut with a pair of micro-surgery scissors. Split-bath experiments with the muscarinic agonist Pilocarpine were conducted as described (*Borgmann et al., 2007*; *2009*). A small, 2 mm stretch of cuticle between the pro- and meso-, and the meso- and metathoracic ganglia was removed except for the connectives, and the gap filled with vaseline (Bad Apotheke, Bad Rothenfelde, Germany). The compartments were first filled with normal *Carausius* saline for control recordings, and the saline in the mesothoracic compartment was subsequently replaced by 3 mM pilocarpine solution in saline.

ML campaniform sensilla (CS) were stimulated according to (*Akay et al., 2007*). One-half to two-thirds of the ML femur were left intact. We mimicked the effect of CS-activation through ML loading during walking by rhythmically bending the femur with a piezoelectric device driven by a ramp generator (both from electronics workshop, University of Cologne) while recording from mesothoracic pro- and retractor MNs in the otherwise deafferented mesothoracic ganglion (*Akay et al., 2007*).

## Data analysis

N denotes the number of animals used for a given condition, n the number of steps evaluated. For comparisons of EMG activity, EMG traces were rectified and smoothed (τ = 50 ms, Spike 2, CED, UK), and each single data point of each step exported to Excel (Microsoft Corp., Seattle, USA) for averaging. For each step, the minimum muscle activity was set to zero and the maximum to 1. In some cases, weak crosstalk from the antagonist muscle was removed by the following procedure: the activity of the EMG in the antagonist and in the agonist were triggered simultaneously (i.e. at lift-off or touchdown of the tarsal contact trace), and exported in the same way as above. Then the antagonists minimum activity was set to 0, but its maximum to an arbitrary value of 0.5, due to the smaller size of the antagonist signal in the agonist EMG. The normalized activity of the antagonistic muscle was then subtracted from the corresponding value of the muscle under investigation. Latencies of the first spike with respect to lift-off or touchdown were calculated relative to the tarsal contact signal (see above). The absolute latency was normalized with respect to the corresponding step cycle and averaged. Average swing/stance phase duration was calculated from each evaluated step from lift-off to touchdown for swing, and from touchdown to lift-off for stance.

For stepping period analysis, we used 14 walks from N=10 animals, the muscle activation profiles during inside and outside were determined from N=5 animals. For the reduced preparations of two-legged animals, we used head movement in addition to leg kinematics to determine the direction of the turn. By using head posture alone and not having four more legs to judge turning behavior from, we may have introduced a bias towards more easily identifiable narrow turns in our sample of the two-leg preparations. However, this has no consequence for the conclusions drawn. For the analysis of the processing of load feedback we analyzed walking sequences from N=13 animals. Data for the analysis of descending influence of two-legged walking on the motor activity of ML pro-/retractor was taken from N=14 to 17 animals depending on the muscle and the direction of the turn (numbers given in the text). For the split-bath experiments, data from N=5 animals for outside, and N=8 animals for inside turns were analyzed. The lesion experiments were performed on N=4 animals (lesion of the pro-mesothoracic connective), and CS stimulation in N=13 animals.

Data processing and figure preparation were performed in Spike2 (Vers. 5.05, CED) and MATLAB 7.0 (The MathWorks, Inc., Natick). In some analyses, extracellular recordings were rectified and smoothed. For EMG analysis, smoothing was performed with the Spike2 smoothing function by calculating the average value of the input data points from time t - T to t + T seconds for each sample point. T was 0.05 s in our analyses. To analyze the activity of the retractor in response to CS stimuli we first rectified and then smoothed the retractor activity with a running Gaussian average (width of 100 ms). For responses during inside steps we then determined the maximum retractor activation in a time window of 150 ms after CS stimulation and normalized these to the average maximum response during control stimulations. The resulting activation strength was then plotted against the instantaneous phase of the FL step cycle. To characterize retractor responses during outside steps we first determined the average retractor activity in a time window of 100 ms before CS stimulation and 100 ms after CS stimulation. We then subtracted the average activity after CS stimulation from the activity before CS stimulation and divided the difference by the sum of the two activities; the resulting value (also known as Michelson contrast) varied between -1 and 1. Values larger than 0 thereby indicated an increase in retractor activation after CS stimulation, values below 0 indicated a decrease. These values were plotted against the phase of the FL step cycle, defined by the start of one stance phase to the start of the next, using circular statistics (*Berens, 2009*). In addition, we determined the instantaneous phase of FL steps during inside steps at the times when either no retractor activation occurred in response to CS stimulation or when protractor activation was observed. For the analysis of the mesothoracic motor activity during FL stepping, data were first analyzed with respect to walking leg step cycle, again defined by the start of one stance phase to the start of the next, using circular statistics (*Berens, 2009*). Phase histograms were used to compare motor neuron activity in steps with different cycle periods. Polar plots for mesothoracic motor activity show mean vectors of activity in the step cycle for each experiment. Vectors that had significant lengths are marked with an asterisk (Rayleigh test with a value of 0.005 [*Batschelet, 1981*]). The vector length from most experiments was highly significant due to the large number of spikes. For the overall mean vector of all experiments no test of significance was done due to the varying number of spikes and steps in the experiments. A cross-correlation between pro- and retractor MN activities

was done for the complete recording time including the time between stepping sequences. Between stepping sequences, pro- and retractor MNs are tonically active (*Büschges and Schmitz, 1991*). The cross-correlation function mirrors the episodic occurrence of the stepping sequences and, if it exists, a periodic coupling between pro- and retractor MN activities. Significance levels marked with asterisks are as follows: * $p<0.05$, ** $p<0.01$, *** $p<0.001$ where applicable. Figures were prepared with Origin (Vers. 8.5, Origin Lab Corp., Northampton, MA) and Photoshop software (Vers.12.0, Adobe Systems Inc., San Jose, CA).

## Acknowledgements

The authors would like to thank Dr. Anke Borgmann for valuable help with the data analysis, and Drs. Joachim Schmidt, Tibor Toth, and Silvia Daun-Gruhn for many valuable comments and input during the study and in the course of the preparation of the manuscript. We would also like to thank Michael Dübbert and Jan Sydow for their excellent technical support. The work was supported by DFG grant Bu857/14 to AB.

## Additional information

### Funding

| Funder | Grant reference number | Author |
|---|---|---|
| Deutsche Forschungsgemeinschaft | Bu857/14 | Ansgar Büschges |

The funders had no role in study design, data collection and interpretation, or the decision to submit the work for publication.

### Author contributions

MG, Conception and design, Acquisition of data, Analysis and interpretation of data, Drafting or revising the article; PR, Acquisition of data, Analysis and interpretation of data, Drafting or revising the article; TB, Analysis and interpretation of data, Drafting or revising the article; AB, Conception and design, Analysis and interpretation of data, Drafting or revising the article

### Author ORCIDs

Matthias Gruhn, http://orcid.org/0000-0003-0115-5189

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
