## [Decision Letter]

Thank you for submitting your work entitled "Body Side-specific Control of Motor Activity during Turning in a Walking Animal" for consideration by *eLife*. Your article has been reviewed by three peer reviewers, and the evaluation has been overseen by Ronald L. Calabrese as the Reviewing Editor and K VijayRaghavan as the Senior Editor.

The following individuals involved in review of your submission have agreed to reveal their identity: Roy Ritzmann and Hans-Joachim Pflüger (peer reviewers).

The reviewers have discussed the reviews with one another and the Reviewing Editor has drafted this decision to help you prepare a revised submission.

Summary:

The authors present a detailed analysis of optomotor-induced turning and the neuronal mechanisms underlying the differences between the leg movements of the two body sides in stick insects. They find that the generation of turning kinematics are the combined result of unilateral commands that change the leg motor output via task-specific modifications in the processing of local sensory feedback as well as modification of the activity of local central pattern generating networks in a body-side-specific way. This work in an important model system for the study of the reflex control of terrestrial limbed locomotion has potential implication for how visually guided turning movements are controlled in terrestrial limbed vertebrates. In this system there is excellent experimental access to reflex mechanisms and to central neurons that integrate reflex inputs and participate in motor pattern generation.

The experiments are carefully done and the data presented is elegant and in general appropriately analyzed. Figures are clear and present the needed data. Writing is clear.

Essential revisions:

There are three main concerns that must be addressed before this paper can be considered for publication, however. The detailed reviews provide the reasoning behind why these changes are essential.

1) The origin of the unilateral commands that are posited to underlie turning movements are not clear. No direct evidence is presented here that these commands descend from the brain and that asymmetric descending activity in connective axons occurs. The authors must clarify this point particularly in light of findings by Martin JP, Guo P, Mu L, Harley CM, Ritzmann RE. (Central-Complex Control of Movement in the Freely Walking Cockroach. Curr Biol. 2015 Nov 2;25(21):2795-803. doi: 10.1016/j.cub.2015.09.044. Epub 2015 Oct 22. PubMed PMID: 26592340.). They should also adjust their claims accordingly.

Alternatively, they can record such asymmetric activity and show that it descends from the brain.

2) How the 'turns' that take place on the frictionless slippery plate relate to turns on a normal substrate is difficult to envision. The authors must provide a compelling discussion of how to relate slippery plate data to normal substrate conditions. Alternatively, new experiments would have to be performed using a more natural situation, e.g., the leg moving on an air suspended ball to make useful comparisons.

3) Reviewer #2 has asked for some new analysis of the existing data which must be performed.

*Reviewer #1:*

This is an interesting study that clearly describes asymmetrical effects on both CPGs and load reflexes associated with turning on inside and outside legs. The authors suggest that such changes are the result of descending commands and provide a model for motor adjustments associated with turning. The data are very nice, but I have some concerns.

First, I am concerned about what they authors mean by "descending effects". They seem to take the correlation of mesothoracic load effects and CPG activation with prothoracic activity as indicating a causative relationship; i.e. the prothoracic ganglion is dictating the changes in mesothoracic. I am not sure that is warranted. It could be that descending commands from higher centers are causing both prothoracic and mesothoracic changes associated with turning in tandem. That would create the correlations that are documented here, but without causation. Moreover, I think the authors need to be careful about what they mean by "descending effects". They performed a nice experiment looking at the effects of unilateral lesion of the T1-T2 connective that indicates a descending effect. But descending from where? This experiment addresses effects from prothoracic to mesothoracic ganglion, but does not address effects from higher centers. In the Discussion, they seem to jump to the brain and in fact to the central complex (CX) (BTW a morphology paper authored by just about all the anatomists who work on insect brain recently agreed to use "CX" as the shorthand for "central complex"). That is entirely possible, but I don't see any data in this paper that addresses influences from the CX or any other brain source. The authors do refer to several CX papers to support that claim. They also describe two systems that have examined effects of brain neurons descending to lower centers in association with turning movements. Interestingly they did not list a recent paper that more directly addresses their hypothesis. Martin et al. (2015) Curr. Biol. 25:1-9 describes motor maps associated with extracellularly recorded CX activity in freely walking cockroaches as they turn and climb over object and that stimulation through the CX electrodes could consistently evoke turns in a particular direction. When that occurred, they moved the preparation, with the brain electrodes in place, to a prep dish that allowed them to test the inter-joint reflexes stemming from the femoral chordotonal organ on the inside leg with and without brain stimulation in the CX region that evoked turns. On the inside leg, Ds reflexes were reversed consistent with what Mu and Ritzmann, 2008 found with bilateral cervical lesion. Another paper that would seem to be relevant is Szczecinski et al. (2014) Biol. Cybern. This paper examined the joint kinematics of all leg joints as a cockroach mounted on an oiled plate transitioned from forward walking to turning in response to an optomoter stimulus. The changes that occurred were then modeled in a neural simulation of joint reflexes and reflexes.

Second, I think the authors need to be careful about what they mean by turning (especially in the Discussion, subsection “Different leg kinematics and motor output on the two body sides of a curve stepping insect”). This paper utilizes only data from oiled plate experiments. I think this preparation is very useful and I agree wholeheartedly with their assertion that the mechanical decoupling of the legs from each other allows them to directly assess central neural effects. I have no problem with that. However, no experimental procedure is without its own baggage and this one has plenty. The animal is slipping and the decoupling could have profound effects on motor control. As the authors point out in the Discussion, some of the differences they report relative to other papers could arise from this. The CPGs could be running at a very high rate that would be limited with a whole suite of inter-leg mechanical coupling. In the intact freely moving insect, the CPGs could be providing a switch of last resort that never actually occurs, because various sensors from various legs normally force the issue before the CPG repolarizes. This is seen nicely in leech heart where neurons on one side of the ganglion will eventually repolarize from a plateau potential but normally are driven to baseline by inhibition from its contralateral homologue long before that occurs in isolation. Moreover, the turning that the authors are reporting on represents just one of many types of turns. The switch to pulling motion that is seen in the inside leg is not always seen in freely walking turns of insects. Jindrich and Full (1999) JEB 202:1603 showed several years ago that many cockroach turns show no kinematic changes but rather result from asymmetrical ground reaction forces. That kind of turn could easily occur simply by a descending command altering the strength of lateral joint movements on one side. The point is, there are many ways that any animal can execute a turn and each could be controlled by different mechanisms.

None of these comments question the relevance of the data that is reported or even the model that is presented. I just think the authors need to be careful to not generalize too much beyond the preparation that they are using and the results that they have to report on. So, I would strongly suggest making it clear that this may or may not relate directly to more natural turning movements, although it probably plays some part in natural turning, and that their data are not addressing commands descending from the brain at this time.

*Reviewer #2:*

This paper analyzes the role of local feedback, descending commands, and the intrinsic CPG on the patterns of antagonist protractor/retractor motor activity on opposing sides of the stick insect during curve walking. The importance of this work lies in its analysis of the contribution of each these factors to differences in the movements of contralateral legs to produce curve walking. Previous work has shown that the inside middle legs move at a lower frequency than the outside legs in animals that are curve walking on normal substrates. In the present experiments, only the front legs contacted the ground, which was frictionless. During frictionless curve walking induced by optokinetic stimuli, the inside legs moved at a higher frequency than the outside legs. The paper describes experiments to determine how the intrinsic left and right CPGs, lateralized descending commands, and imposed load stimuli each affect the inside and outside leg motor patterns during curve walking. While this analysis is clear, its relevance to the different motor patterns seen during curve-walking on normal substrates is not made clear. As the authors discuss, "Most studies on turning of invertebrates also reported a decrease in stepping frequency on the inside compared to the outside legs (Zolotov et al., 1975; Jander, 1985), sometimes no change in frequencies (Cruse and Saavedra, 1996)". Under these conditions, the legs are coupled through movements of the body relative to the substrate. This coupling is absent during frictionless walking. The authors suggest that "The higher inside stepping frequency under these (frictionless substrate) conditions may therefore reflect true centrally generated drive during turning." But the higher stepping may also reflect the faster movements of the leg and therefore a more frequent generation of feedback signals that promote stance/swing and swing/stance transitions. Indeed, the factors analyzed under frictionless walking may operate differently to produce the different walking patterns observed on normal substrates. The authors should show how the mechanisms at work during frictionless curve walking inform our understanding of curve walking on normal substrates.

In the second paragraph of the subsection “Contribution of load feedback to turning kinematics”: Figure 2 does not present evidence that the load response "was independent of the phase of the stimulation with respect to the FL steps." Bii shows when the stimulus was presented with respect to the FL step, and Biii appears to show a cumulative PSTH for all the responses. A plot of the response amplitude vs. phase of the FL step is needed to show the phase independence of the response.

In the second paragraph of the subsection “Contribution of load feedback to turning kinematics”: To show that "the observed motor effects are independent of FL stepping (Figure 3Bii)", the authors will need a plot of the response amplitude vs. phase of the step. Some stimuli evoked increases and some evoked decreases in retractor activity; the authors should show how these vary across the phase of the step.

*Reviewer #3:*

The manuscript by Gruhn et al. describes experiments in stick insects that examine the central aspects of turning behaviour under the interesting condition where the animal performs stationary turns on a "slippery surface". The main finding is that inside and outside legs receive different inputs from their ipsilateral (side specific) connective which not only alters the involved central pattern generators but also the processing of local sensory feedback. Not totally unexpected from discussions and theories of previous work, it is the experimental evidence presented here, that supports this idea very convincingly. Therefore, I do not have any objections to the scientific content of this manuscript. In general, experiments and results are clearly described and documented. Of course, additionally one really would like to see recordings from the respective connectives and whether their patterns differ on both sides and whether it would be possible to identify some of the descending neurons which would be causal for the observed effects on inside and outside legs.

---

## [Author Response]

Essential revisions:

There are three main concerns that must be addressed before this paper can be considered for publication, however. The detailed reviews provide the reasoning behind why these changes are essential.

1) The origin of the unilateral commands that are posited to underlie turning movements are not clear. No direct evidence is presented here that these commands descend from the brain and that asymmetric descending activity in connective axons occurs. The authors must clarify this point particularly in light of findings by Martin JP, Guo P, Mu L, Harley CM, Ritzmann RE. (Central-Complex Control of Movement in the Freely Walking Cockroach. Curr Biol. 2015 Nov 2;25(21):2795-803. doi: 10.1016/j.cub.2015.09.044. Epub 2015 Oct 22. PubMed PMID: 26592340.). They should also adjust their claims accordingly.

Alternatively, they can record such asymmetric activity and show that it descends from the brain.

2) How the 'turns' that take place on the frictionless slippery plate relate to turns on a normal substrate is difficult to envision. The authors must provide a compelling discussion of how to relate slippery plate data to normal substrate conditions. Alternatively, new experiments would have to be performed using a more natural situation, e.g., the leg moving on an air suspended ball to make useful comparisons.

3) Reviewer #2 has asked for some new analysis of the existing data which must be performed.

First of all we would like to thank you for the many critical and helpful comments and criticism that you have raised. We have tried to answer to them as well as we could. We believe that we have improved the manuscript considerably, and hopefully to your satisfaction. We have also added some more analyses that were requested and modified Figure 2 and Figure 3 accordingly, as well as additional supplements to these two figures. In the following, we have interwoven our answers to the comments raised by each reviewer into your comments.

Reviewer #1:

This is an interesting study that clearly describes asymmetrical effects on both CPGs and load reflexes associated with turning on inside and outside legs. The authors suggest that such changes are the result of descending commands and provide a model for motor adjustments associated with turning. The data are very nice, but I have some concerns.

First, I am concerned about what they authors mean by "descending effects". They seem to take the correlation of mesothoracic load effects and CPG activation with prothoracic activity as indicating a causative relationship; i.e. the prothoracic ganglion is dictating the changes in mesothoracic. I am not sure that is warranted. It could be that descending commands from higher centers are causing both prothoracic and mesothoracic changes associated with turning in tandem. That would create the correlations that are documented here, but without causation. Moreover, I think the authors need to be careful about what they mean by "descending effects". They performed a nice experiment looking at the effects of unilateral lesion of the T1-T2 connective that indicates a descending effect. But descending from where? This experiment addresses effects from prothoracic to mesothoracic ganglion, but does not address effects from higher centers. In the Discussion, they seem to jump to the brain and in fact to the central complex (CX) (BTW a morphology paper authored by just about all the anatomists who work on insect brain recently agreed to use "CX" as the shorthand for "central complex"). That is entirely possible, but I don't see any data in this paper that addresses influences from the CX or any other brain source. The authors do refer to several CX papers to support that claim. They also describe two systems that have examined effects of brain neurons descending to lower centers in association with turning movements. Interestingly they did not list a recent paper that more directly addresses their hypothesis. Martin et al. (2015) Curr. Biol. 25:1-9 describes motor maps associated with extracellularly recorded CX activity in freely walking cockroaches as they turn and climb over object and that stimulation through the CX electrodes could consistently evoke turns in a particular direction. When that occurred, they moved the preparation, with the brain electrodes in place, to a prep dish that allowed them to test the inter-joint reflexes stemming from the femoral chordotonal organ on the inside leg with and without brain stimulation in the CX region that evoked turns. On the inside leg, Ds reflexes were reversed consistent with what Mu and Ritzmann, 2008 found with bilateral cervical lesion. Another paper that would seem to be relevant is Szczecinski et al. (2014) Biol. Cybern. This paper examined the joint kinematics of all leg joints as a cockroach mounted on an oiled plate transitioned from forward walking to turning in response to an optomoter stimulus. The changes that occurred were then modeled in a neural simulation of joint reflexes and reflexes.

If the reviewer understands our claims this way, we apologize. We have not meant "descending effects" to mean that the prothoracic ganglion is dictating changes in the mesothoracic ganglion. In fact, we do believe that the inputs causing the changes in the mesothoracic motor activity and the processing of mesothoracic sensory inputs is caused by descending influence from the brain and may be influenced by prothoracic activity under certain circumstances. In order to take the reviewers very valid concerns into account we have reworded the entire manuscript carefully in order not to suggest such an effect and pay tribute to the very valid concerns. For example, we have removed the term "descending" from the final paragraph of the Introduction and carefully reworded conclusions in the Results part (see below). In addition, we have integrated the citations the reviewer mentions into the manuscript as we feel that especially Martin et al. work strongly supports our claims. This influential work had clearly skipped our attention in the final phase of pre-submission. We have also carefully reworded the entire Discussion with regard to the origin of the "descending" inputs. In addition, we have modified Figure 7 such that it is more clear that the arrows within the scheme only suggest the minimum origin for descending inputs we are sure about (i.e. anterior to the mesothorax or FL influence during inside steps) and broken lines indicate in our view likely but untested origin from the brain.

*Second, I think the authors need to be careful about what they mean by turning (especially in the Discussion, subsection “Different leg kinematics and motor output on the two body sides of a curve stepping insect”). This paper utilizes only data from oiled plate experiments. I think this preparation is very useful and I agree wholeheartedly with their assertion that the mechanical decoupling of the legs from each other allows them to directly assess central neural effects. I have no problem with that. However, no experimental procedure is without its own baggage and this one has plenty. The animal is slipping and the decoupling could have profound effects on motor control. As the authors point out in the Discussion, some of the differences they report relative to other papers could arise from this. The CPGs could be running at a very high rate that would be limited with a whole suite of inter-leg mechanical coupling. In the intact freely moving insect, the CPGs could be providing a switch of last resort that never actually occurs, because various sensors from various legs normally force the issue before the CPG repolarizes. This is seen nicely in leech heart where neurons on one side of the ganglion will eventually repolarize from a plateau potential but normally are driven to baseline by inhibition from its contralateral homologue long before that occurs in isolation. Moreover, the turning that the authors are reporting on represents just one of many types of turns. The switch to pulling motion that is seen in the inside leg is not always seen in freely walking turns of insects. Jindrich and Full (1999) JEB 202:1603 showed several years ago that many cockroach turns show no kinematic changes but rather result from asymmetrical ground reaction forces. That kind of turn could easily occur simply by a descending command altering the strength of lateral joint movements on one side. The point is, there are many ways that any animal can execute a turn and each could be controlled by different mechanisms.*

None of these comments question the relevance of the data that is reported or even the model that is presented. I just think the authors need to be careful to not generalize too much beyond the preparation that they are using and the results that they have to report on. So, I would strongly suggest making it clear that this may or may not relate directly to more natural turning movements, although it probably plays some part in natural turning, and that their data are not addressing commands descending from the brain at this time.

The reviewer is absolutely right in his call for caution in the use of the term "turning" and we have re-written the Discussion thoroughly to make sure potential readers will be aware of the advantages and disadvantages of the approach that we have used, and the implied caveats. Especially the paragraph on "Different leg kinematics and motor output on the two body sides of a curve stepping insect" has been thoroughly edited in this respect.

Reviewer #2:

This paper analyzes the role of local feedback, descending commands, and the intrinsic CPG on the patterns of antagonist protractor/retractor motor activity on opposing sides of the stick insect during curve walking. The importance of this work lies in its analysis of the contribution of each these factors to differences in the movements of contralateral legs to produce curve walking. Previous work has shown that the inside middle legs move at a lower frequency than the outside legs in animals that are curve walking on normal substrates. In the present experiments, only the front legs contacted the ground, which was frictionless. During frictionless curve walking induced by optokinetic stimuli, the inside legs moved at a higher frequency than the outside legs. The paper describes experiments to determine how the intrinsic left and right CPGs, lateralized descending commands, and imposed load stimuli each affect the inside and outside leg motor patterns during curve walking. While this analysis is clear, its relevance to the different motor patterns seen during curve-walking on normal substrates is not made clear. As the authors discuss, "Most studies on turning of invertebrates also reported a decrease in stepping frequency on the inside compared to the outside legs (Zolotov et al., 1975; Jander, 1985), sometimes no change in frequencies (Cruse and Saavedra, 1996)". Under these conditions, the legs are coupled through movements of the body relative to the substrate. This coupling is absent during frictionless walking. The authors suggest that "The higher inside stepping frequency under these (frictionless substrate) conditions may therefore reflect true centrally generated drive during turning." But the higher stepping may also reflect the faster movements of the leg and therefore a more frequent generation of feedback signals that promote stance/swing and swing/stance transitions. Indeed, the factors analyzed under frictionless walking may operate differently to produce the different walking patterns observed on normal substrates. The authors should show how the mechanisms at work during frictionless curve walking inform our understanding of curve walking on normal substrates.

We have tried to make the analysis of frictionless walking in comparison to normal walking more clear. While the reviewer is right that the higher inside stepping frequency may reflect faster movement of the leg due to the frictionless walking conditions, this does not explain while this should be different on the outside and not create a higher outside frequency. Therefore, we still believe that the inherent drive between inside and outside is different. However, we have re-worded the text carefully to take the concern into account. It now reads: "A plausible explanation for the difference to our findings and similar ones in the cockroach (Mu et al., 2005) is that we studied curve walking on a slippery surface. […] However, because the slipperiness takes away the influences of friction and reduces the sensory feedback between legs, the significantly higher stepping frequency on the inside compared to the outside, under these conditions may reflect true centrally generated drive during turning."

In the second paragraph of the subsection “Contribution of load feedback to turning kinematics”: Figure 2 does not present evidence that the load response "was independent of the phase of the stimulation with respect to the FL steps." Bii shows when the stimulus was presented with respect to the FL step, and Biii appears to show a cumulative PSTH for all the responses. A plot of the response amplitude vs. phase of the FL step is needed to show the phase independence of the response.

The reviewer is right that our plot of the response to CS stimuli during outside steps of the ipsilateral FL does show phase but absolute timing of the CS stimulus. We phrased this incorrectly. We have also further analyzed the phasing of the response and modified the text accordingly. In Figure 2, we added a plot with the retractor activity vs CS stimulus with respect to phase. In addition, we added a Figure 2—figure supplement 1 with the data from all 13 animals.

In the second paragraph of the subsection “Contribution of load feedback to turning kinematics”: To show that "the observed motor effects are independent of FL stepping (Figure 3Bii)", the authors will need a plot of the response amplitude vs. phase of the step. Some stimuli evoked increases and some evoked decreases in retractor activity; the authors should show how these vary across the phase of the step.

The reviewer is absolutely correct in that there is variability in the strength of the retractor activation during inside steps of the FL. We have re-analyzed the retractor response to load stimuli during inside turns to evaluate the response amplitude with respect to phase. We found that there is no correlation between the response amplitude and the phase of the FL step cycle during which the stimulus occurred. This is shown as a plot in Figure 3. In addition, we added Figure 3—figure supplement 1 with the data from all 13 animals. Furthermore, we have added phase response plots for protractor activation and the failure to respond in 3E and F.

Reviewer #3:

The manuscript by Gruhn et al. describes experiments in stick insects that examine the central aspects of turning behaviour under the interesting condition where the animal performs stationary turns on a "slippery surface". The main finding is that inside and outside legs receive different inputs from their ipsilateral (side specific) connective which not only alters the involved central pattern generators but also the processing of local sensory feedback. Not totally unexpected from discussions and theories of previous work, it is the experimental evidence presented here, that supports this idea very convincingly. Therefore, I do not have any objections to the scientific content of this manuscript. In general, experiments and results are clearly described and documented. Of course, additionally one really would like to see recordings from the respective connectives and whether their patterns differ on both sides and whether it would be possible to identify some of the descending neurons which would be causal for the observed effects on inside and outside legs.

We thank the reviewer for this very good suggestion and agree that recordings from the connectives would be a very nice addition to the data presented here. Plans to conduct these experiments are underway.